# Beyond Myopia: Learning from Positive and Unlabeled Data through Holistic Predictive Trends

**Xinrui Wang**[*]   **Wenhai Wan**[*]   **Chuanxing Geng**   **Shaoyuan Li**[†]   **Songcan Chen**[†]

College of Computer Science and Technology, Nanjing University of Aeronautics and Astronautics
MIIT Key Laboratory of Pattern Analysis and Machine Intelligence

## Abstract

Learning binary classifiers from positive and unlabeled data (PUL) is vital in many real-world applications, especially when verifying negative examples is difficult. Despite the impressive empirical performance of recent PUL methods, challenges like accumulated errors and increased estimation bias persist due to the absence of negative labels. In this paper, we unveil an intriguing yet long-overlooked observation in PUL: *resampling the positive data in each training iteration to ensure a balanced distribution between positive and unlabeled examples results in strong early-stage performance. Furthermore, predictive trends for positive and negative classes display distinctly different patterns.* Specifically, the scores (output probability) of unlabeled negative examples consistently decrease, while those of unlabeled positive examples show largely chaotic trends. Instead of focusing on classification within individual time frames, we innovatively adopt a holistic approach, interpreting the scores of each example as a temporal point process (TPP). This reformulates the core problem of PUL as recognizing trends in these scores. We then propose a novel TPP-inspired measure for trend detection and prove its asymptotic unbiasedness in predicting changes. Notably, our method accomplishes PUL without requiring additional parameter tuning or prior assumptions, offering an alternative perspective for tackling this problem. Extensive experiments verify the superiority of our method, particularly in a highly imbalanced real-world setting, where it achieves improvements of up to $11.3\%$ in key metrics. The code is available at https://github.com/wxr99/HolisticPU.

## 1   Introduction

Positive and Unlabeled Learning (PUL) is a binary classification task that involves limited positive labeled data and a large amount of unlabeled data [36]. This learning scenario naturally arises in many real-world applications like matrix completion[25], deceptive reviews detection[45], fraud detection[35] and medical diagnosis[56]. It also serves as a key component of more complex machine learning problems, such as out-of-distribution detection[63] and adversarial training[18]. Two main categories of PUL methods are cost-sensitive methods and sample-selection methods. However, both approaches face their challenges. The cost-sensitive methods rely on the negativity assumption, which may introduce estimation bias due to the mislabeling of positive examples as negative[49]. This bias can be accumulated and even worsen during later training stages, making its elimination challenging. The sample-selection methods struggle with distinguishing reliable negative examples, particularly during the initial stage, which also results in error accumulation during the training process[23; 57].

As a basic component for various PUL methods, resampling the positive labeled data shows its potential in alleviating the bias brought by negative assumption [49; 52; 30; 33; 61]. For example,

---

[*]Equal contribution: Xinrui Wang <wangxinrui@nuaa.edu.cn> and Wenhai Wan <wwh35@nuaa.edu.cn>
[†]Corresponding authors: Songcan Chen <s.chen@nuaa.edu.cn> and Shaoyuan Li <lisy@nuaa.edu.cn>.

37th Conference on Neural Information Processing Systems (NeurIPS 2023).

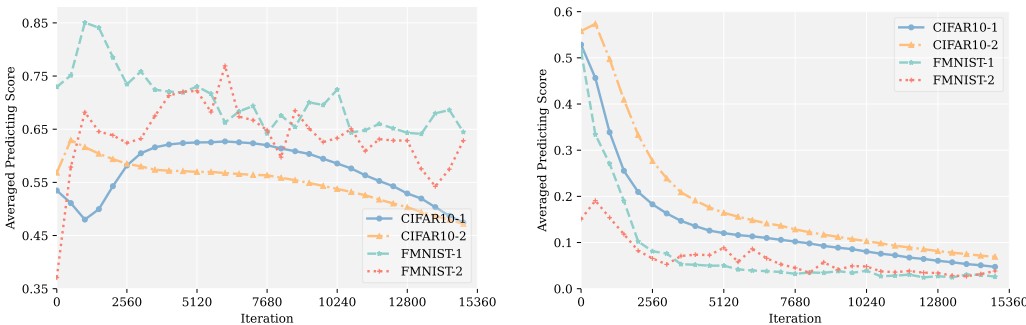

Figure 1: Averaged predicting scores (output probability) of positive (left) and negative (right) examples in an unlabeled dataset during the first 15,360 iterations of training (30 epochs).

[30] resamples positive examples according to the given class prior and assumed label mechanism to achieve decent performance. In this paper, we dive deeper into this class of strategies. Instead of relying on one single-step prediction which is prone to model uncertainty, we take a holistic view and examine the predictive trend of unlabeled data during the training process. Specifically, we treat the unlabeled data as negative. In each training epoch, we resample over the labeled positive data to ensure a balanced class distribution. We evaluate the model's performance on CIFAR10 and FMNIST datasets[32; 55] with 4 experimental settings. Our pilot experiments show that this resampling method achieves comparable or even state-of-the-art test performance at the outset, but underperforms soon after. Furthermore, the averaged predicting scores (output probability) of unlabeled negative examples exhibit a consistent decrease, whereas those of unlabeled positive examples display an initial increase before subsequent decreasing or oscillating. Conclusively, the averaged predictive trends for different classes exhibit significant differences, as depicted in Figure 1. One possible explanation for these observations is the model's early focus on learning simpler patterns, which aligns with the early learning theory of noisy labels [37]. Although the resampling strategy enjoys these advantages, selecting an appropriate model can be more challenging than the classification task itself due to the lack of a precise validation set.

To break the above limitation, we propose a novel approach that treats the predicting scores of each unlabeled training example as a temporal point process (TPP). It takes a holistic view and surpasses existing methods that focus on examining loss values or tuning confidence thresholds based on a limited history of predictions. By centering on the difference in trends of predicting scores, our approach provides a more comprehensive understanding of deep neural network training in PUL. To further investigate whether this difference in trends is prevalent in individual unlabeled examples, we apply the Mann-Kendall Test, a non-parametric statistical test used to detect trends in the temporal point process [20], to the continuously predicting scores of each example. These scores are classified into three types: *Decreasing*, *Increasing*, and *No Trend*. The statistical test reveals a clear distinction in the trends of predicted scores for each positive and negative example, supporting our observation. Our findings suggest that utilizing the model's classification ability in the early stages may be sufficient for successfully classifying unlabeled examples. This discovery offers us a new perspective on reformulating the problem of distinguishing positive and negative examples in the unlabeled set as identification of their corresponding predictive trends.

We then propose a novel TPP-inspired measure, called **trend score** to quantify the distinctions in predictive trends. It is obtained by applying a robust mean estimator [3] to the expected value of the ordered difference in a TPP (sequence of predicting scores for each example)[19]. Subsequently, we introduce a modified version of Fisher's Natural Break to distinguish these predictive trends, identifying a natural break point in the distribution of **trend score**. This approach divides examples into two groups: the group with **high trend score** represents positive examples, while the group with **low trend score** corresponds to negative examples. Our approach simplifies the training process by circumventing threshold selection when assigning pseudo-labels. Once the unlabeled data is classified, the remaining problem becomes a binary supervised learning task, and issues such as estimating class priors can be easily addressed. In summary, our main contributions are:

- We demonstrate the effectiveness of the proposed resampling strategy. It is also observed that predictive trends for each example can serve as an important metric for discriminating the categories of unlabeled data, providing a novel perspective for PUL.

- We propose a new measure, **trend score**, which is proved to be asymptotically unbiased in the change of predicting scores. We then introduce a modified version of Fisher's Natural Break with lower time complexity to identify statistically significant partitions. This process does not require additional tuning efforts and prior assumptions.

- We evaluate our proposed method with various state-of-the-art approaches to confirm its superiority. Our method also achieves a significant performance improvement in a highly imbalanced real-world setting.

## 2 Our Intuition and Method

### 2.1 Preliminary

We first revisit some important notations in PUL. Formally, let $x \in \mathbb{R}^d$ be the input data with $d$ dimensions and $y \in \{0, 1\}$ be the corresponding label. Different from the traditional binary classification, PUL dataset is composed of a positive set $\mathcal{P} = \{x_i, y_i = 0\}_{i=1}^{n_p}$ and an unlabeled set $\mathcal{U} = \{x_i\}_{i=1}^{n_u}$, where the unlabeled set $\mathcal{U}$ contains both positive and negative data. Throughout the paper, we denote the positive class prior as $\pi = \mathbb{P}(y = 0)$.

### 2.2 Resampling Strategies for Positive and Unlabeled Learning

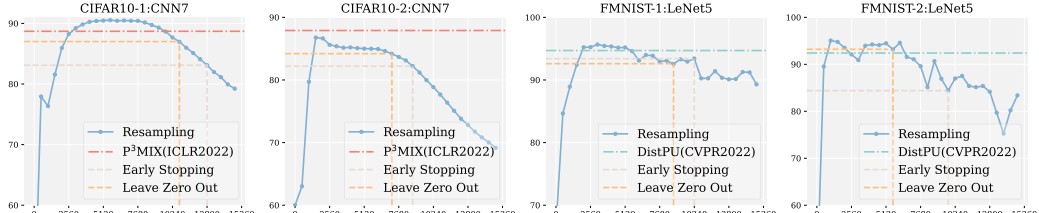

Figure 2: The accuracy of our resampling method (first 30 epochs). The horizontal line represents the accuracy of the state-of-the-art methods. Early stopping and Leave Zero Out represent different model selection strategies.

Resampling strategies have long been a baseline for dealing with imbalanced data or limited labels, which naturally fits PUL since its key challenge lies in limited labels and potentially imbalanced data distribution[5]. Different from popular resampling strategies applied in PUL[30], we follow the training scheme as [47; 58] to independently sample positive and unlabeled data as different data batches and the loss function is defined accordingly.

$$\mathcal{L} = \frac{1}{|\mathcal{B}p|} \sum_{(x_i, y_i) \in \mathcal{B}_p} \ell(\hat{y}_i, y_i) + \frac{1}{|\mathcal{B}u|} \sum_{x_i \in \mathcal{B}_u} \ell(\hat{y}_i, 1), \quad \hat{y}_i = f(x_i). \tag{1}$$

Here, we denote $f \in \mathcal{F}$ as a binary classifier, $\ell(\cdot, \cdot)$ as the loss function, $\mathcal{B}_p$ and $\mathcal{B}_u$ as the positive and unlabeled training batches respectively. We ensure that $|\mathcal{B}_p| = |\mathcal{B}_u|$ to achieve a balanced class prior during the training process. This approach emphasizes the labeled data and mitigates the imbalance of positive and pseudo-negative labels, which also provides a good theoretical explanation when dealing with high-dimensional data conforming to different Gaussian distributions. As shown in Appendix A.1, an optimal decision hyperplane can be attained when $|\mathcal{P}|/|\mathcal{U}|$ equals 1. Figure 2 details the performance of our resampling baseline on two datasets under four different settings. It can be observed that the proposed method performs comparably or even better than state-of-the-art methods (P³MIX[33] and DistPU[61]) in the early stages of training, as demonstrated by its test performance at certain epochs. However, the method's performance quickly degrades in all 4 settings as the estimation bias worsens during training due to the false negatives introduced by the negativity assumption. We also explore alternative model selection strategies, such as holding out a validation set from given labeled examples or using different versions of augmented data for model selection, as inspired by prior studies [34; 39]. In addition to the common practice of selecting the model from an additional positive validation set, we also implement LZO[34], which selects the model based on the

mixup-induced validation set. As shown in Table 1, the performance gap persists, especially when most of the unlabeled data belongs to the positive class.

Table 1: Classification accuracy (Recall rate is reported on Credit Card) on unlabeled training data. Resampling-P represents the model selected on an extra positive validation set. Resampling-LZO represents the model selected through LZO. Resampling* represents the best model selected on the test set which is an ideal case.

| Dataset | F-MNIST-1 | F-MNIST-2 | CIFAR10-1 | CIFAR10-2 | STL10-1 | STL10-2 | Credit Card | Alzheimer |
|---|---|---|---|---|---|---|---|---|
| Resampling-P | 89.93 | 84.29 | 81.06 | 72.93 | - | - | 60.75 | 70.09 |
| Resampling-LZO | 93.37 | 92.04 | 84.87 | 82.98 | - | - | 67.24 | 74.11 |
| Resampling* | 94.92 | 94.57 | 89.56 | 85.46 | - | - | 87.54 | 76.30 |
| P$^3$MIX-C | 91.59 | 87.65 | 86.05 | 88.14 | - | - | 76.21 | 68.01 |

To tackle the above issues, some denoising-based semi-supervised PUL methods, such as [8; 52; 49], have leveraged some threshold tuning or sample selection techniques to achieve acceptable empirical performance. These techniques have been criticized in [54] for relying solely on prediction scores or loss values, as they do not account for uncertainty in the selection process. This becomes even more problematic in PUL, where the noise ratio is typically higher when making a negativity assumption[2].

To break the above limitations, we record the whole predicting process of each unlabeled training example to take a holistic view of the training. It is evident that averaged model-predicting scores for positive and negative data display two distinct trends when implementing the above resampling strategy in the early training stages. Meanwhile, the standard deviation of predictions for positive examples increases rapidly during training, making it increasingly difficult to select an appropriate threshold for distinguishing between positive and negative examples. The appropriate threshold interval for discriminating positive and negative examples quickly shrinks as training progresses, indicating that existing denoising techniques cannot fundamentally alleviate the issues of accumulated errors and increased estimation bias. Therefore, a more robust evaluation measure is necessary beyond relying on raw model-predicted scores or loss values. Implementation details in model selection and visualizations of threshold tuning are provided in Appendix A.

## 2.3 Identifying Predictive Trends: A Key to Successful Classification

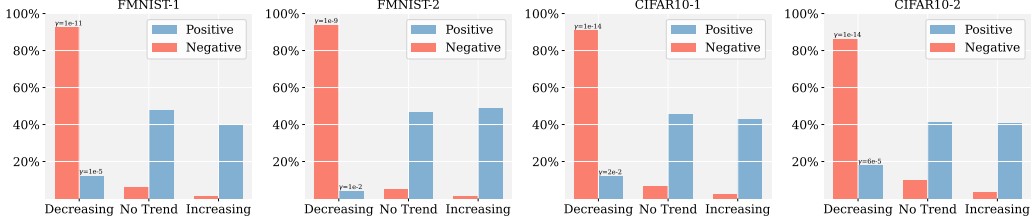

Figure 3: The Mann-Kendall Test is performed on 4 settings of CIFAR10 and FashionMnist datasets. The figure reports the fractions of positive and negative examples in an unlabeled dataset exhibiting different predictive trends during the early training stage (first 30 epochs).

While deep neural networks have strong learning capabilities, they are at risk of overfitting all provided labels, regardless of their correctness. This can result in all unlabeled examples being predicted as negative [1; 59]. We expect the predictive scores of negative examples in the unlabeled set to consistently decrease because all negative examples are given true negative labels by the negativity assumption. On the other hand, the predictive scores of positive examples in the unlabeled training set may not decrease initially because the resampled labeled examples are consistently emphasized from the start of training. To provide more evidence, we use the Mann-Kendall test to analyze the model-predicted scores of each example [20]. This test categorizes the prediction sequence into three situations: *Decreasing*, *Increasing*, and *No Trend*. The calculation process of the Mann-Kendall Test is detailed in Appendix B. Figure 3 shows a contrast between the trends of predicted scores for positive and negative examples. Even when certain positive and negative examples exhibit a similar trend of decreasing prediction scores during training, we observed significant differences in the significance index $\gamma$ across different classes.

Our next objective is to measure the differences between positive and negative examples. To accomplish this, we require an evaluation measure that captures the significance of the observed trends in model-predicted scores. Before developing our own measure, an important notation in the TPP is first introduced, $\mathbb{E}[\Delta p]$, which represents the expected value of the ordered difference in a series of predicting scores.

$$\mathbb{E}[\Delta p] = \lim_{t \to \infty} \frac{2}{t(t-1)} \sum_{i<j}^{t} \Delta p_{ij}, \ \Delta p_{ij} = p_j - p_i. \tag{2}$$

where $p_i$ is the predicting score (output probability) at $i$-th epoch, $t$ is the number of training epochs.

$$\tilde{S} = \frac{2}{t(t-1)} \sum_{i=1}^{t-1} \sum_{j=i+1}^{t} \Delta p_{ij}, \ \Delta p_{ij} = p_j - p_i. \tag{3}$$

While $\tilde{S}$ is the empirical mean and unbiased estimation of $\mathbb{E}[\Delta p]$, it can be unreliable for non-Gaussian examples and may not handle outliers or heavy-tailed data distributions well as illustrated in[3]. To address these issues, we propose a robust mean estimator inspired by[54; 20], called the **trend score** $S$, which measures the difference between each ordered pair of prediction scores:

$$\hat{S} = \frac{2}{t(t-1)} \sum_{i=1}^{t-1} \sum_{j=i+1}^{t} \psi(\alpha \Delta p_{ij}), \ \Delta p_{ij} = p_j - p_i. \tag{4}$$

$$\psi(\Delta p_{ij}) = sign(\Delta p_{ij}) \cdot log(1 + |\Delta p_{ij}| + \Delta p_{ij}^2/2). \tag{5}$$

in which $\alpha > 0$ is a scaling parameter, and $sign()$ is the sign function that returns $-1$ if its argument is negative, 0 if its argument is zero, and 1 if its argument is positive. The function $\psi(\cdot)$ can result in a more robust estimation by flattening the values of $\Delta p_{ij}$ and reducing the influence of minority outlier points on the overall estimation. Besides, we also provide a simplified version as:

$$\dot{S} = \frac{1}{t-1} \sum_{i=1}^{t-1} \psi(\alpha \Delta p_{ij}), \ \Delta p_{ij} = p_j - p_i. \tag{6}$$

Notably, $\tilde{S}, \hat{S}, \dot{S}$ are all calculated on each example. Experiments show that both $\hat{S}, \dot{S}$ exhibit better empirical results than $\tilde{S}$ in Section3. For choosing the stopping epoch $t$, we implement the LZO[34] algorithm as described in Section2.2. We also derive a concentration inequality between our **trend score** $\hat{S}$ and the expected value of the ordered difference $\mathbb{E}[\Delta p]$.

**Theorem 2.1.** *Let $P = \{p_{ij}|1 \leq i \leq t-1, 2 \leq j \leq t, i < j\}$ be an observation set of changes in predictions in which $\mathbb{E}[\Delta p]$ is the expected values of the ordered difference in a temporal point process and $\sigma^2$ is the variance of $P$. By exploiting the non-decreasing influence function $\psi(\cdot)$, for any $\epsilon > 0$, we have the following bound with probability at least $1 - 2\epsilon$:*

$$|\hat{S} - \alpha \mathbb{E}[\Delta p]| < \frac{2\alpha\sigma\sqrt{\frac{2log(\epsilon^{-1})}{t(t-1)}}}{1 - \sqrt{\frac{2log(\epsilon^{-1})}{t(t-1)\alpha^2\sigma^2}}} = O\left((log(\epsilon^{-1}))^{\frac{1}{2}} t^{-1}\right). \tag{7}$$

It illustrates that the measure we propose is an asymptotically unbiased estimation with a linear weighting of $\mathbb{E}[\Delta p]$. The proof is provided in AppendixC. It is also proved in [3] that the deviations of this robust mean estimator can be of the same order as the deviations of the empirical mean computed from a Gaussian statistical sample, which further verifies the advantage of this estimator.

## 2.4 Clustering Unlabeled Data by the Fisher Criterion

The topic of accurately labeling unlabeled data is widely discussed in various fields, including PUL. In the existing literature, threshold-based criteria and small loss criteria are the two primary approaches used for selecting reliable or clean examples, as seen in studies such as [47; 58; 29; 49]. However, previous works generally select examples based solely on current predictions, ignoring the inherent uncertainty in training examples, leading to longer training times and poor generalization ability[54; 41]. Besides, they often require extensive hyperparameter tuning efforts to choose

appropriate thresholds or ratios for data selection. In this section, we introduce a new labeling approach based on our proposed **trend score** tackling the above issues.

Our proposed **trend score** is the naturally comparable one-dimensional data and allows the Fisher Criterion to be a viable choice. It identifies a natural break point in the trend score distribution, which could be used to divide the data into two groups: one with high trend scores and one with low trend scores representing positive and negative examples respectively. Specifically, the objective function of finding this Fisher's natural break point can be formed as follows:

$$\min_{C_1,C_2} \frac{\sum_{x \in C_1}(\hat{S}_x - \mu_1)^2}{|C_1|} + \frac{\sum_{x \in C_2}(\hat{S}_x - \mu_2)^2}{|C_2|} \tag{8}$$
$$s.t. \ C_1 \cap C_2 = \emptyset, \ C_1 \cup C_2 = x_1, x_2, \ldots, x_N.$$

where $\hat{S}_x$ is our derived **trend score** for example $x$, $C_1$ and $C_2$ are the two clusters, $\mu_i$ is the mean of cluster $C_i$, and $N$ is the total number of data points. We utilize the Fisher natural break point method to automatically determine a threshold value that divided the trend score distribution into two distinct groups. Our implementation introduces an improved algorithm, which reduces the time complexity from $O(N^2)$ to $O(Nlog(N))$, as explained in Appendix D. This method eliminates the need for manual threshold selection or hyperparameter tuning, both of which can be time-consuming and error-prone. Furthermore, the data-driven approach we used optimizes the threshold value for the specific dataset under analysis, rather than relying on arbitrary or pre-defined values.

Once the unlabeled data is classified, the remaining task becomes a straightforward supervised learning problem. We directly train by a cross-entropy loss on the estimated labels given by Eq.8 on the backbone network given in Table 4. Besides, issues such as estimating class priors can be addressed easily when unlabeled data are classified.

## 3 Experiments

### 3.1 Classification on Unlabeled Training Set

In this subsection, we first evaluate the performance of our method on the unlabeled training set compared with some state-of-the-art methods. As shown in Table 2, our method demonstrates excellent classification performance on the unlabeled training data (the true labels of unlabeled data are not available in STL10). Moreover, a comparison with state-of-the-art prior estimation methods in PUL is conducted to further verify the effectiveness of our approach, and the results are presented in Table 3.

Table 2: Classification accuracy (Recall rate is reported on Credit Card) on unlabeled training data.

| Dataset | F-MNIST-1 | F-MNIST-2 | CIFAR10-1 | CIFAR10-2 | STL10-1 | STL10-2 | Credit Card | Alzheimer |
|---------|-----------|-----------|-----------|-----------|---------|---------|-------------|-----------|
| nnPU | 85.31 | 82.46 | 83.11 | 83.23 | - | - | 62.53 | 64.01 |
| PGPU | 92.02 | 90.17 | 85.67 | 88.38 | - | - | 42.12 | 75.09 |
| Self-PU | 94.04 | 91.59 | 84.06 | 83.77 | - | - | 71.00 | 70.05 |
| $P^3$MIX-C | 91.59 | 87.65 | 86.05 | 88.14 | - | - | 76.21 | 68.01 |
| Ours | **95.41** | **96.00** | **91.42** | **91.17** | - | - | **98.90** | **75.13** |

Table 3: Absolute estimation error with the true positive prior in the first row. We implement an oracle early stopping for the extant methods as defined in [15]. Our method significantly reduces estimation error when compared with existing methods.

| Algorithm | F-MNIST-1 | F-MNIST-2 | CIFAR10-1 | CIFAR10-2 | STL10-1 | STL10-2 | Credit Card | Alzheimer |
|-----------|-----------|-----------|-----------|-----------|---------|---------|-------------|-----------|
| $\pi$ | 0.40 | 0.60 | 0.40 | 0.60 | 0.50 | 0.50 | 0.05 | 0.50 |
| KM2 | 0.146 | 0.106 | 0.115 | 0.164 | 0.096 | 0.101 | 0.236 | 0.094 |
| BBE* | 0.082 | 0.073 | 0.034 | 0.059 | 0.046 | 0.064 | 0.112 | 0.026 |
| $(TED)^n$ | 0.026 | 0.020 | 0.042 | 0.044 | 0.024 | 0.021 | 0.018 | 0.014 |
| Ours | **0.014** | **0.021** | **0.016** | **0.031** | **0.018** | **0.009** | **0.004** | **0.011** |

### 3.2 Test Performance

We use three synthetic prevalent benchmark datasets including FashionMnist (F-MNIST) [55], CIFAR10 [32] and STL10 [10] and two real-world datasets on fraud detection[1] and Alzheimer

---
[1]https://www.kaggle.com/datasets/mlg-ulb/creditcardfraud

Table 4: Dataset description and corresponding backbones.

| Dataset | #Trainset | #Testset | Input size | Backbone |
|---------|-----------|----------|------------|----------|
| F-MNIST | 60,000 | 10,000 | $28\times28$ | LeNet-5 |
| CIFAR-10 | 50,000 | 10,000 | $3\times32\times32$ | 7-Layer CNN |
| STL-10 | 105,000 | 8,000 | $3\times96\times96$ | 7-Layer CNN |
| Alzheimer | 5,890 | 1,279 | $3\times224\times224$ | ResNet-50 |
| Credit Fraud | 8,392 | 2098 | 30 | 6-Layer MLP |

diagnosis[2] as our test set. We provide the dataset description and corresponding backbones in Table4, and the positive priors of each setting are given in Table3. More detailed description of benchmark datasets, dataset split and implementation details are given in AppendixF. For each dataset, we run our method for 5 times with different random seeds and report the averaged classification accuracy. We follow the settings in [52; 61] when making the comparison: randomly select 769 positive examples in Alzheimer dataset, 100 positive examples in Credit Fraud dataset and 1000 positive examples in others as the labeled set in training. Classification accuracy on test sets is reported as the main criterion. For highly imbalanced distributed (Credit Fraud) and biasedly selected (Alzheimer) datasets, we provide additional metrics such as Recall, F1 score and AUC on test sets for a more comprehensive comparison.

Table 5: Results of classification accuracy (%) on 3 generic datasets with 6 settings (mean±std).

| Algorithm | F-MNIST-1 | F-MNIST-2 | CIFAR10-1 | CIFAR10-2 | STL10-1 | STL10-2 |
|-----------|-----------|-----------|-----------|-----------|---------|---------|
| uPU | 81.6±1.2 | 85.7±2.6 | 76.5±2.5 | 71.6±1.4 | 76.7±3.8 | 78.2±4.1 |
| nnPU | 91.4±0.6 | 90.2±0.7 | 84.7±2.4 | 83.7±0.6 | 77.1±4.5 | 80.4±2.7 |
| Self-PU | 90.8±0.4 | 89.1±0.7 | 85.1±0.8 | 83.9±2.6 | 78.5±1.1 | 80.8±2.1 |
| PAN | 87.7±2.4 | 89.9±3.2 | 87.0±0.3 | 82.8±1.0 | 77.7±2.5 | 79.8±1.4 |
| vPU | 92.6±1.2 | 90.5±0.8 | 86.8±1.2 | 82.5±1.1 | 78.4±1.1 | 82.9±0.7 |
| MIXPUL | 90.4±1.2 | 89.6±1.2 | 87.0±1.9 | 87.0±1.1 | 77.8±0.7 | 78.9±1.9 |
| PULNS | 91.0±0.5 | 89.1±0.8 | 87.2±0.6 | 83.7±2.9 | 80.2±0.8 | 83.6±0.7 |
| Dist-PU | 94.7±0.4 | 92.4±0.4 | 86.8±0.7 | 87.2±0.9 | 79.8±0.6 | 82.9±0.4 |
| $P^3$MIX-E | 92.6±0.4 | 91.8±0.2 | 88.2±0.4 | 84.7±0.5 | 80.2±0.9 | 83.7±0.7 |
| $P^3$MIX-C | 92.8±0.6 | 90.4±0.1 | 88.7±0.4 | 87.9±0.5 | 80.7±0.7 | 84.1±0.3 |
| Ours | **95.8±0.3** | **96.0±0.3** | **91.1±0.2** | **90.3±0.1** | **83.7±0.3** | **85.3±0.6** |

### 3.2.1 Sythetic datasets

Our proposed method consistently outperforms all PUL baselines by 1% to 4% on all generic benchmark datasets and settings, as shown in Table 5, demonstrating its superior performance. Furthermore, many existing PUL methods rely on a given positive prior or make various assumptions that are not available in real-world settings, whereas our method does not require any of them. To avoid inherent challenges such as accumulated errors and estimation bias, we transform the above challenges into a much simpler task of discerning the trend of the model-predicting scores. Considering we can achieve outstanding classification accuracy in unlabeled data, it is natural to expect our method to outperform existing PUL methods. While using some tricks for label noise learning like Co-teaching[22] and large loss criterion[28] could possibly further improve the performance of our method, we believe that in most scenarios, our method can effectively solve existing PUL problems with simplicity.

### 3.2.2 Real-world datasets

This subsection presents experimental results on two real-world datasets, including one highly imbalanced Credit Fraud dataset. In fraud detection, recall is typically more important than precision or accuracy, as the consequences of missing a fraudulent transaction can be much more severe than flagging a legitimate transaction as fraudulent. As shown in Table 6, our proposed method achieves significantly higher recall rates and F1 scores, as well as comparable accuracy and precision, indicating its ability to better handle highly imbalanced scenarios. Our approach offers a novel perspective compared to traditional prediction-based methods, as the model's predictive trends are not affected by the positive prior, as long as the observation outlined in Section 2.3 holds. Furthermore,

---

[2]https://www.kaggle.com/datasets/tourist55/alzheimers-dataset-4-class-of-images

Table 6: Comparative results(%) on Credit Card Fraud dataset (mean±std).

| Algorithm | F1 score | Recall | Accuracy | Precision | AUC |
|---|---|---|---|---|---|
| uPU | 89.5±3.1 | 83.4±1.3 | 97.0±0.2 | 96.5±3.6 | 93.4±3.1 |
| nnPU | 89.9±1.0 | 83.4±1.3 | 98.4±0.1 | 97.4±1.1 | 94.2±0.9 |
| nnPU+mixup | 89.0±2.8 | 82.9±1.6 | 98.1±0.1 | 96.0±3.2 | 93.8±2.9 |
| Self-PU | 89.0±2.4 | 85.8±2.0 | 99.2±0.1 | 92.4±3.4 | 95.6±2.8 |
| PAN | 91.5±0.9 | 85.4±1.3 | **99.1±0.1** | 98.5±1.0 | 96.6±1.1 |
| VPU | 91.7±3.9 | 84.9±5.7 | 98.6±0.5 | **99.7±0.6** | 96.9±3.1 |
| MIXPUL | 82.9±2.8 | 86.6±1.3 | 98.4±0.3 | 79.2±3.5 | 91.3±0.7 |
| PULNS | 89.0±2.0 | 83.2±2.1 | 99.0±0.1 | 95.6±1.9 | 94.5±0.7 |
| Dist-PU | 87.9±3.4 | 80.2±4.1 | 98.8±0.4 | 97.2±1.6 | 96.5±2.7 |
| P$^3$MIX-E | 91.9±2.1 | 87.7±2.0 | 99.0±0.1 | 96.5±1.8 | 97.5±0.9 |
| P$^3$MIX-C | 90.2±1.4 | 86.5±1.8 | 98.8±0.1 | 94.1±1.2 | 97.3±1.2 |
| Our Method | **99.1±0.2** | **99.0±0.2** | **99.1±0.1** | 99.3±0.1 | **99.7±0.1** |

Table 7: Comparative results(%) on Alzheimer dataset (mean±std).

| Algorithm | F1 score | Recall | Accuracy | Precision | AUC |
|---|---|---|---|---|---|
| uPU | 67.6±2.8 | 66.1±6.1 | 68.5±2.2 | 69.7±3.5 | 73.8±2.9 |
| nnPU | 68.6±3.2 | 69.5±7.2 | 68.3±2.1 | 68.0±2.3 | 72.9±2.8 |
| RP | 62.1±5.6 | 64.6±15.9 | 61.6±3.2 | 61.9±4.5 | 66.1±3.3 |
| PUSB | 69.2±2.4 | 69.3±2.4 | 69.2±2.4 | 69.2±2.4 | 74.4±2.4 |
| PUbN | 70.4±3.2 | 72.0±8.4 | 70.0±1.3 | 69.4±2.5 | 70.0±1.3 |
| Self-PU | 72.1±1.1 | 75.4±5.1 | 70.9±0.7 | 69.3±2.5 | 75.9±1.8 |
| aPU | 70.5±3.4 | 75.7±8.2 | 68.5±1.8 | 66.2±0.9 | 70.7±3.7 |
| VPU | 70.2±1.1 | 76.7±3.6 | 67.4±0.7 | 64.7±1.1 | 73.1±0.9 |
| ImbPU | 68.8±1.9 | 70.6±6.5 | 68.2±0.8 | 67.5±2.5 | 73.8±0.7 |
| Dist-PU | 73.7±1.6 | **80.1±5.1** | 71.6±0.6 | 68.5±1.2 | **77.1±0.7** |
| Our Method | **74.5±2.4** | 79.5±5.8 | **72.8±0.9** | **70.2±1.6** | **77.1±2.3** |

our method also demonstrates comparable performance on the Alzheimer dataset to the state-of-the-art method DistPU, which employs various regularization techniques and data augmentation strategies. In both two real-world settings, our method achieves a balanced good performance on all evaluation metrics which further illustrates its effectiveness.

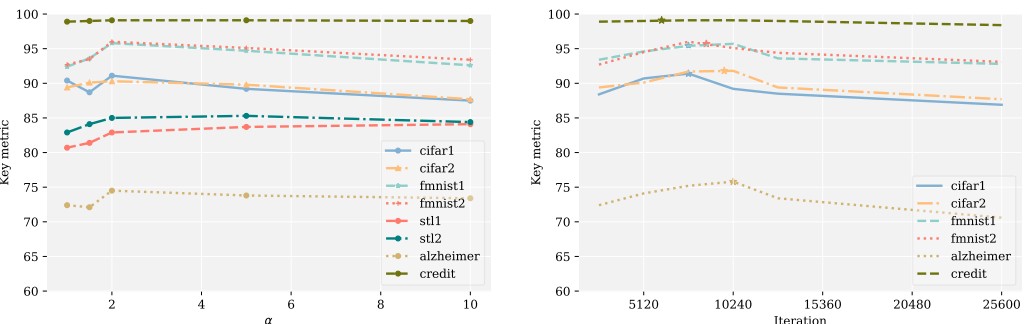

Figure 4: Sensitivity analysis was performed on two parameters: $\alpha$ (left) and stopping iteration (right). The stopping iteration of LZO (also the one we use) is denoted by '$*$' on the right.

### 3.2.3 Ablation Study

To investigate the specific effects of different components (Resampling, **trend score**, and Fisher Natural Break Partition) in our method, we conducted a series of ablation studies and compared them with some popular alternatives. From Table 8, we can draw several observations: (1) The resampling strategy plays a crucial role in our method as it maximizes the discrepancy of the trends in different classes of examples, particularly in the Credit Fraud dataset. It serves as an important factor in amplifying the model's early success, which is the foundation of our further approach

Table 8: Ablation results (%) on CIFAR-10 (acc), Credit Fraud (recall) and Alzheimer (f1 score). "✓" indicates the enabling of the corresponding components.

| | | Trend Measure | | | Clustering | | Dataset | | |
|---|---|---|---|---|---|---|---|---|---|
| Resampling | | TS | Simplified TS | MK | Natural break | k-means | CIFAR10-1 | Credit Fraud | Alzheimer |
| | | ✓ | | | ✓ | | 84.1 | 88.6 | 69.2 |
| ✓ | | ✓ | | | | ✓ | 89.4 | 99.3 | 70.5 |
| ✓ | | | | ✓ | ✓ | | 90.2 | 99.0 | 69.7 |
| ✓ | | | ✓ | | ✓ | | 90.7 | 99.2 | 73.9 |
| ✓ | | ✓ | | | ✓ | | 91.1 | 99.1 | 74.5 |

towards achieving better performance. (2) Our proposed **trend score** provides a better evaluation metric than the statistic $\tilde{S}$ used in the standardized Mann-Kendall test, and the simplified **trend score** also shows competitive performance. (3) Fisher Natural Break Partition derives deterministic optimal partitions with better statistical properties and empirical performance compared to heuristic k-means. Moreover, it is unrelated to initialization and less time-consuming than the original version, as detailed in AppendixD.

### 3.2.4 Sensitivity Analysis

In this subsection, we investigate the impact of two hyperparameters, namely the scaling parameter $\alpha$ and the stopping iteration (we do not need to manually tune it), on the evaluation of predictive trends for each example. To facilitate comparisons, we set $\alpha$ to 2 and employ the LZO algorithm [34] discussed in Section 2.2 for selecting the stopping epoch in our experiments involving mixed labeled data. As depicted in Figure 4, our approach consistently delivers robust outcomes across diverse hyperparameter values. Moreover, the model tends to perform better when $\alpha > 1$ and demonstrates basically consistent performance. Figure 4 confirms the effectiveness of the LZO strategy which is free of manual intervention in the stopping epoch.

## 4   Related Works

For a long time, learning with limited supervision has been a striking task in the machine learning community and PUL is an emerging paradigm of weakly supervised learning [64; 17]. Despite its close relations with some similar concepts, the term PUL is generally accepted from [36; 12; 14]. Currently, the mainstream PUL methods cast this problem as a cost-sensitive classification task through importance reweighting, among which uPU [13] is the widely known one. Later, the authors of nnPU [31] suggest that uPU gets overfitting when using flexible and complex models such as Deep Neural Networks and thus propose a non-negative risk estimator. Some recent studies attempt to combine the cost-sensitive method with model's capability to calibrate and distill the labeled set with various techniques like denoise [49], self-paced curriculum [8] and heuristic mix up [33; 52].

Parallel with the cost-sensitive methods, another branch of PUL methods adopts a heuristic two-step method. The early trials of two-step methods mainly focus on the sample-selection task to form a reliable negative set and further yield the semi-supervised learning framework [57; 35; 23; 6; 27]. Other two-step methods are mainly derived from the large margin principle to correct the bias caused by unreliable negative data such as Loss Decomposition [46], Large margin based calibration and label disambiguation [16; 60]. Plus, different techniques have been employed to assign labels for unlabeled data in PUL like Graph-based models [4; 62], GAN [24; 27] and Reinforcement learning [38] in recent years. Plus, decision tree based PU methods are also investigated in [53].

Most PUL methods are oriented from a SCAR (selected completely at random) assumption or established on a given class prior. In this respect, there emerges some class prior estimation algorithms specially designed for PUL. PE attempts to minimize the Pearson divergence between the labeled and unlabeled distribution, PEN-L1 [9] and MPE [15] are then proposed to modify PE by using a simple Best Bin Estimation (BBE) technique. Unfortunately, most class prior estimation algorithms still rely on specific assumptions and the estimates will be unreliable otherwise[40]. Regarding the possibility of selection bias in the labeling process, the SCAR assumption is relaxed in [30]. VAE-PU is the first generative PUL model without a supposed labeling mechanism like SCAR assumption [42] and further investigated in [51]. For more details about PUL, readers are referred to a recent survey for a comprehensive understanding of this subject [2].

# 5    Conclusion

This study introduces a novel method for Positive-Unlabeled Learning (PUL) that takes a fresh perspective by identifying the unique characteristics of each example's predictive trend. Our approach is based on two key observations: Firstly, resampling positive examples to create a balanced training distribution can achieve comparable or even superior performance to existing state-of-the-art methods in the early stages of training. Secondly, the predicting scores of negative examples tend to exhibit a consistent decrease, while those of positive examples may initially increase before ultimately decreasing or oscillating. These insights lead us to reframe the central challenge of PUL as a task of discerning the trend of the model predicting scores. We also propose a novel labeling approach that uses statistical methods to identify significant partitions, circumventing the need for manual intervention in determining confidence thresholds or selecting ratios. Extensive empirical studies demonstrate the effectiveness of our method and its potential to contribute to related fields, such as learning from noisy labels and semi-supervised learning.

# 6    Acknowledgments and Disclosure of Funding

This work was supported by the Natural Science Foundation of China (NSFC) (Grant No.62376126), the National Key R&D Program of China (2022ZD0114801), National Natural Science Foundation of China (61906089), Natural Science Foundation of China (NSFC) (Grant No.62106102), Natural Science Foundation of Jiangsu Province (BK20210292), Graduate Research and Practical Innovation Program at Nanjing University of Aeronautics and Astronautics (xcxjh20221601).

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
