# OpenReview forum: "Beyond Myopia: Learning from Positive and Unlabeled Data through Holistic Predictive Trends"
_NeurIPS.cc/2023/Conference — NeurIPS 2023 spotlight_

### Official Review · Reviewer_3ytg · 2023-06-30

**Soundness:** 3 good
**Presentation:** 3 good
**Contribution:** 3 good
**Rating:** 7
**Confidence:** 4

**Summary:**

In this paper, the authors investigate a phenomenon appearing in positive and unlabeled data learning. i.e. predictive trends is different for positive and negative classes in binary classification task. Based on this observation, the authors propose a holistic approach to capture the predictive trends of different instances. Extensive experiments on real-world and synthetic data sets validate the effectiveness of proposed method.

**Strengths:**

(1) The phenomenon appearing in positive and unlabeled data learning is interesting, i.e. predictive trends is different for positive and negative classes in binary classification task.
(2) Extensive experiments on real-world and synthetic data sets validate the effectiveness of proposed method.

**Weaknesses:**

However, there still exists several shortcomings to be overcome as follows:
(1) In experiments, the comparing algorithms are not presented.
(2) In Figure 3, what does it mean "No trend"? The trend score is a constant or not?
(3) By using trend score, the problem of unlabeled data classification is transformed as threshold selection, which requires that the trend score exhibits obvious difference for these two classes.  Although the authors leverage Fisher Criterion to induce the positive and negative classes, as shown in Figure 1, the predictive trend on CIFAR10 is not completely consistent with the assumption. How does the author view this issue?

=====after rebuttal=====
The authors' responses have well addressed most of my concerns of this paper. I would like to raise my overall score from 6 to 7 and recommend acceptance on this paper.

**Questions:**

By using trend score, the problem of unlabeled data classification is transformed as threshold selection, which requires that the trend score exhibits obvious difference for these two classes.  Although the authors leverage Fisher Criterion to induce the positive and negative classes, as shown in Figure 1, the predictive trend on CIFAR10 is not completely consistent with the assumption. How does the author view this issue?

=====after rebuttal=====
The authors' responses have well addressed most of my concerns of this paper. I would like to raise my overall score from 6 to 7 and recommend acceptance on this paper.

**Limitations:**

Yes

---

> ### Author Rebuttal · Authors · 2023-08-07
>
> Thanks for your insightful comments. We address your concerns as follows.
>
> ### Q1: Missing comparing algorithms in experiments
> Thank you for your valuable suggestions. Actually, due to space limitations, we have included the comparing algorithms in Section F of the Appendix. We will also add a brief introduction of the comparing methods in the experiments section.
> ### Q2: Explanation of 'No trend'
> We apologize for any confusion that may have occurred. It is important to clarify that the Mann-Kendall Trend Test is a statistical test used to analyze the presence of a monotonic trend in a chronological series of a variable. The test can yield three possible results: 'Increasing' if there is a significant increasing trend, 'Decreasing' if there is a significant decreasing trend, or 'No trend' if there is no significant monotonic tendency observed.
>
> In some cases, there may be chronological series that do not exhibit a clear increasing or decreasing trend. These series are categorized as having 'No trend'. It's worth noting that the Mann-Kendall Trend Test is primarily focused on detecting monotonic trends rather than more complex patterns.
>
> Additionally, the trend score is calculated to represent the predictive trend of each example. Once the phase 1 training is completed, the trend score can be considered a constant value for that particular example.
> ### Q3: Predictive trend on CIFAR10 is not completely consistent with the assumption.
> We acknowledge that the assumptions regarding the predictive trend of unlabeled positive examples may not hold true for all cases. However, based on the statistical results shown in Figure 3 for CIFAR, it appears that the majority of examples are consistent with our assumption.
>
> Even in cases where unlabeled positive examples exhibit a decreasing trend, the significance index $\gamma$ demonstrates a substantial difference between the positive and negative classes. This suggests that the proposed trend score is still able to distinguish between the two classes effectively, despite the specific trend direction.
>
> It's important to consider when the assumption may not hold universally, so the proposed trend score adopts a more robust mean estimator such that the overall statistical results and differences in significance indices support the reliability of our method. If you have any more questions or need further clarification, please let us know.

---

### Official Review · Reviewer_EzoK · 2023-07-03

**Soundness:** 3 good
**Presentation:** 3 good
**Contribution:** 3 good
**Rating:** 7
**Confidence:** 4

**Summary:**

The paper focuses on addressing the positive and unlabeled learning problem. The authors make an observation that a simple resampling method can yield strong early-stage performance, which has been overlooked in previous literature. Building on this insight, the paper proposes a trend detection measure to tackle the PU learning problem. The proposed method is supported by both theoretical and empirical results.

**Strengths:**

1. This work studies an important problem, and it is well-written and easy to follow.
2. The motivation and proposed method are clearly described. It's definitely a new way to approach the positive and unlabeled learning problem from the perspective of trend detection.
3. The proposed PU learning approach requires no explicit computation of the class prior and avoids the need for additional tuning efforts and assumptions.
4. The experiments are well-conducted, and comprehensively compared to recent SOTA methods. Ablations and sensitivity analysis are also abundant.

Overall, the paper presents a novel insight for the positive and unlabeled learning problem , and the contribution is definitely relevant and significant.

**Weaknesses:**

1. Although it is mentioned that the proposal can be applied to more machine learning topics, further explanation should be included.
2. Can other clustering methods could yield better empirical performance compared with Algorithm 1

**Questions:**

1. In addition to the positive and unlabeled learning problem, can the proposed approach be applied to other machine learning topics such as semi-supervised learning and learning with noisy labels
2.Why does the performance of the proposed approach vary with different choices of positive priors For example, in Tab10, when highly imbalanced positive priors are used, the proposed algorithm displays a notable gap in performance.
1. Can the proposed Fisher natural break point method be considered a universal approach for multi-class classification, or are there limitations to its applicability

**Limitations:**

This is an algorithmic work, and the authors have addressed some limitations.

---

> ### Author Rebuttal · Authors · 2023-08-07
>
> Thanks for your insightful comments. We address your concerns as follows.
>
> ### Q1: Further discussions on how this method applies to other areas
> Please kindly refer to our responses to Question 3 from Reviewer vUs8.
> ### Q2: Whether the method can perform well with different positive priors?
> As shown in Tab10, our method achieves very competitive performance under various positive priors compared with methods specialized for imbalanced cases. We admit that there still exists a performance gap between different positive priors. This gap may be explained by both the upper bounds and the lower bounds mentioned in Lemma G.1 and G.2. However, we maintain our belief that the predictive trends derived from our proposed resampling method can be a viable choice for the general imbalanced scenarios.
>
> ### Q3: Is Fisher natural break point a universal approach?
>  Yes, the Fisher natural break point is a universal approach that can be used to find any number of break points and can be applied to multi-class classification tasks. For a detailed exploration, please refer to:
>
>  Jenks, George F. "The data model concept in statistical mapping." International yearbook of cartography 7 (1967): 186-190.
>
>  Jiang, Bin. "Head/tail breaks: A new classification scheme for data with a heavy-tailed distribution." The Professional Geographer 65.3 (2013): 482-494.
>
> ### Q4: Can other clustering methods achieve better performance?
> As demonstrated in Table 8, the Fisher break point consistently outperforms the traditional k-means algorithm across the majority of datasets. Moreover, the Fisher Natural Break Partition generates deterministic optimal partitions with superior statistical properties compared to many existing clustering methods that heavily rely on heuristics. Additionally, this method is independent of initialization and requires less computational time, thanks to its time complexity of $O(nlogn)$.

---

> > ### Comment · Reviewer_EzoK · 2023-08-20
> >
> > The authors answer all my concerns, and I keep my initial score on this work.

---

### Official Review · Reviewer_3dQ9 · 2023-07-07

**Soundness:** 3 good
**Presentation:** 3 good
**Contribution:** 3 good
**Rating:** 6
**Confidence:** 4

**Summary:**

In this paper, the authors investigate the topic of positive and unlabeled learning, and solve it from an interesting perspective, namely trend score. The story begins with the empirical observations of different temporal output score trends of positive instances, unlabeled negative instances, and unlabeled positive instances. Based on the observations, the authors propose to compute the trend score under the temporal point process, and use the trend scores to distinguish positive and negative from unlabeled instances. The authors conduct numbers of experiments, and the results show the effectiveness of the proposed method.

**Strengths:**

 The idea is novel for PU learning.
- The paper is easy-to-follow.
- Extensive empirical results.


**Weaknesses:**

Please refer to the following Questions section.

**Questions:**

 Overall, I argue the idea of this paper is interesting, but I still have several questions:
1.	If the trend score is the major contribution, whether the resampling is a must in the framework? What are the relationships between them?
2.	The authors compute the trend scores for all unlabeled instances and then use them to identify the positive instances from them. Is that right?
3.	In the problem formulation, the positive label is denoted as y=0. So why the predictive trend of positive instances is “increasing”?

---

> ### Author Rebuttal · Authors · 2023-08-07
>
> Thanks for your insightful comments. We address your concerns as follows.
>
> ### Q1: Whether resampling is a must in the framework? What's the relationship between trend score and resampling?
>
> We argue that resampling is essential in this framework. As we emphasized in Sections 2.2 and 2.3, a benign early learning performance serves as the foundation for our later observations on trend score. Additionally, resampling also amplifies the differences in trend score predictions between examples from different classes.
>
> ### Q2: Is it right to compute the trend scores for all unlabeled instances?
> Indeed, we calculate trend scores for all unlabeled instances to enhance the differentiation between positive and negative classes. This process is similar to a typical threshold selection procedure, where we compare the trend score of each instance with the derived threshold to make the classification.
>
> ### Q3: Why the predictive trend of positive instances is “increasing”?
>
> We apologize for any confusion caused. The predicting scores represent the probability of an example belonging to the positive class: $P(y=0)$. Therefore, the predictive trend of positive examples does increase in the early stages.

---

### Official Review · Reviewer_vUs8 · 2023-07-08

**Soundness:** 3 good
**Presentation:** 3 good
**Contribution:** 3 good
**Rating:** 7
**Confidence:** 4

**Summary:**

The paper addresses the positive and unlabeled learning problem and presents interesting and meaningful observations. Inspired by the observations, the authors propose an innovative approach that effectively discriminates between positive and negative examples within unlabeled data. The performance of this approach is validated on both synthetic and real-world datasets, further confirming its superiority.


**Strengths:**

- (Clarity) The paper is well organized and clearly written,  the figures are also informative and well-designed.
- (Novelty) The paper is thought-provoking! Handling the PUL problem from the perspective of predicting trends is highly innovative. Furthermore, the proposed techniques are also novel as far as I can tell.
- (Quality) The paper is of good quality in my opinion. The algorithm is well designed for identifying trend prediction (via Trend Score) and clustering unlabeled data (via the improved Fisher Criterion). The technical details are all correct as far as I can tell. The empirical evaluation is also comprehensive, covering 6 datasets, including synthetic and real-world datasets, and comparing against popular baselines.
- (Significance) The proposed techniques are simple, easy to implement and experimentally highly effective, making the algorithm a strong, potentially impactful baseline for future researchers & practitioners to use and/or improve upon.

**Weaknesses:**

1. It would be appropriate to discuss why there can be significantly different prediction trends for positive and negative examples in unlabeled data.
2. Both Table 1 and Table 2 lack the experiments on STL10, although you mentioned in Table 2 that the reason is "the true labels of unlabeled data are not available in STL10." However, a better choice would have been to provide an explanation for the absence of this experiment from the first occurrence.

**Questions:**

1. Is it feasible to apply trend prediction methods in other fields, such as semi-supervised learning?
2. Does recording the predicted values at every iteration for trend analysis yield the best performance? What if we record the predictions every 3 iterations instead? how about every 5 iterations?

---

> ### Author Rebuttal · Authors · 2023-08-07
>
> Thanks for your insightful comments. We address your concerns as follows.
>
> ### Q1: Why prediction trends for positive and negative examples are significantly different?
>
> The divergence in trends between positive and negative examples can primarily be attributed to the strong fitting capabilities of deep neural networks (DNNs). This often leads to the prediction of all unlabeled examples as negative. However, due to the consistent emphasis placed on resampled labeled examples, positive examples exhibit discernible trends in their predictive scores that differ from those of negative examples.
>
> ### Q2: Table 1 and Table 2 lack the experiments on STL10
>
> We sincerely apologize for any confusion caused and appreciate your valuable suggestions. In order to address the absence of this experiment from the initial entry in Table 1, we will provide a detailed explanation.
>
> ### Q3: Is it feasible to apply trend prediction methods in other fields?
>
> In relation to this question, we are inclined to provide a positive response. We can offer intuitive and straightforward approaches for leveraging out-of-distribution (OOD) data detection or semi-supervised learning, such as:
> - In the context of semi-supervised learning, we can consider unlabeled data as the "U" in PUL (Positive-Unlabeled Learning). By utilizing our proposed trend score, we can identify unlabeled examples that belong to a specific class. This enables the formation of multiple binary classifiers for a multi-class classification task.
> - Similarly, we can utilize the predictive trend of a deep neural network (DNN) to detect OOD data. By analyzing the trend exhibited by the DNN's predictions, we can effectively identify instances that lie outside the distribution of the training data.
>
> Overall, we firmly believe that our proposed method can serve as a valuable component within more intricate machine learning techniques.
>
> ### Q4: Does the recording frequency affect the model's performance?
>
> Here, we provide some further experiments on the influence of recording frequency on the model's performance.
>
> | Frequency   | CIFAR10-1   | CIFAR10-2 | FMNIST10-1 | FMNIST10-2 |
> | :---:       |    :----:   |      :---:| :---:      | :---:      |
> | 1           | 91.1 $\pm$ 0.2     | 90.3  $\pm$ 0.1 | 95.8 $\pm$  0.3   | 96.0 $\pm$  0.3   |
> | 3           | 90.9 $\pm$  0.2    | 90.4 $\pm$ 0.3  | 94.8 $\pm$  0.2   | 95.5 $\pm$  0.2   |
> | 5           | 90.6 $\pm$  0.5    | 90.0 $\pm$ 0.3  | 95.1 $\pm$  0.4   | 95.7 $\pm$  0.4   |
>
> Our observations indicate that the recording frequency (capturing predictions at intervals of 1/3/5 iterations) does not appear to have a substantial influence on the performance of the model.

---

> > ### Comment · Reviewer_vUs8 · 2023-08-20
> >
> > Thanks for the author's response. I would also maintain my initial score of accept.

---

### Official Review · Reviewer_cdNW · 2023-07-09

**Soundness:** 3 good
**Presentation:** 3 good
**Contribution:** 3 good
**Rating:** 7
**Confidence:** 4

**Summary:**

The paper proposed an interesting idea of using the trend detection technique for positive and unlabeled learning (PUL). The core problem of PUL is thus reformulated as recognizing trends in these scores. A new TPP-inspired measure is then utilized to solve the task. The idea of the paper is interesting; it is well-motivated and well-supported with a range of experiments from fraud detection and computer vision tasks.

-----------------------
After rebuttal: I am satisfied with the rebuttal and would like to increase my score.

**Strengths:**

1. The proposed technique is well motivated and clearly distinguished from prior works.
2. The PUL model achieves good performance in the early training stage is an interesting phenomenon and worth studying.
3. The proposed method is technically sound and it demonstrates a strong performance against both two-step methods and popular reweighting strategies. Plus, comprehensive experimental results on benchmark datasets clearly demonstrate the phenonmenon and the effectiveness of the proposed method.

**Weaknesses:**

1. The paper attributes the performance downgrade after the early stage success to the label noise problem and the data imbalance problem. However, it is unclear whether all existing reweighting or resampling methods can achieve the same state-of-the-art (SOTA) test set performance in the early training stage and witness a quick performance degradation.
2. std should be included in the experimental results such as those demonstrated in Figure 1.

**Questions:**

1. The comparison with ImbalancedPU in Table 10 in Appendix is interesting to me. Can authors provide a more detailed experimental setting? As the proposed technique is clearly distinguished from prior works. Can we expect the method also achieves decent performance under a class-conditional label noise or imbalance? Some reference:
   1. "Centroid Estimation with Guaranteed Efficiency: A General Framework for Weakly Supervised Learning"
   2. "Learning from Corrupted Binary Labels via Class-Probability Estimation"
2. It's mentioned that the proposed approach does not rely on the "selected completely at random" (SCAR) assumption? Some further discussions are suggested.
3. What does stopping iteration mean in Sec3?

**Limitations:**

Yes

---

> ### Author Rebuttal · Authors · 2023-08-07
>
> # Reviewer cdNW:
>
> Thanks for your insightful comments. We address your concerns as follows.
>
> ### Q1: It is unclear whether existing methods witness the same phenomenon.
>
> A： Most PUL methods currently use a combination of regularization techniques and refinement on mislabeled examples brought by the negativity assumption. To investigate the early learning phenomenon of some recent methods during the training process, we compared their performance in Tables 1 and 2. Our observations suggest that some methods, such as Self-PU and PGPU, perform better than others but are similar to vanilla resampling strategies. Therefore, we opted for the simpler and more commonly used technique of resampling strategies to effectively showcase our discoveries.
> ### Q2: std should be included in Figure1
>
> A: Thanks for your advice, it will be added in the revised version.
>
> ### Q3: What does stopping iteration mean in Sec3?
>
> A: It represents when should we stop the phase 1 training. Concretely, it equals the *MaxEpoch* in **Algorithm 2**.
>
> ### Q4: Detailed experimental setting & extention on class-conditional label noise
>
> A: For a detailed description of our experimental settings, we kindly request the reviewer to refer to Section F in the Appendix. As for the extension on class-conditional label noise, we have included the relevant settings in Table 10. In this case, positive labels are randomly assigned while the class prior is highly imbalanced. Although learning from corrupted binary labels is an interesting problem and highly related to our topic, it is beyond the scope of the PUL and we will talk about it in related works and leave it for our future work.
>
> ### Q5: Further discussions on method's reliance on the SCAR assumption
> Unlike traditional reweighting methods that rely on assumptions to derive an unbiased risk estimator, our method takes a new perspective to tackle PUL. By considering the existence of the early learning phenomenon, the proposed trend score can be perceived as a criterion to differentiate positive and negative data even without the SCAR assumption.

---

### Comment · Area_Chair_92T6 · 2023-08-16
**Author-Reviewer Discussion Period Closing Soon**

Thank you reviewers for your work in evaluating this submission, and thank you authors for responding to the reviewers’ questions and concerns. We are entering the final phase of the discussion period, which will run until August 21st, and some of the authors' responses have to been acknowledged by all the reviewers.

Reviewers: If you have any lingering questions or comments on the rebuttal or the responses, now is the time to express them. At the very least, please acknowledge that you have read the authors’ response to your review.

Thank you everyone for making the review process a fruitful, constructive, and civil process.

AC

---

### Decision · Program_Chairs · 2023-09-21

**Decision:**

Accept (spotlight)

**Comment:**

This paper studies positive and unlabeled learning (PU learning) by observing that predictive trends for positive and negative classes are distinct. Motivated by this, the authors propose a holistic method based on TPP to detect the trend and classify the unlabeled data. The proposed method is novel and verified by empirical studies.
All reviewers agree with the acceptance of this paper. During the discussion stage, the authors provided responses to the reviewers and the reviewers maintained their initial score. Therefore, I recommend the acceptance.